Development and evaluation of a deep learning model for occlusion classification in intraoral photographs

Zhang Rongxiu
Zhang Lin
Zhang Di
Wang Ying
Huang Yongsong
Wang Dong
Xu Li kqxuli@163.com
Department of Stomatology, The First Affiliated Hospital of Bengbu Medical College , Bengbu, Anhui , China
Tribst João Paulo
Electronic publication date: 2025 Sep 26
Publication date: 2025
Volume: 13
Electronic Location ID: e20140
Received 2025 Mar 27; Accepted 2025 Sep 4
Copyright: ©2025 Zhang et al.
Copyright year: 2025
Copyright holder: Zhang et al.
License: This is an open access article distributed under the terms of the Creative Commons Attribution License, which permits unrestricted use, distribution, reproduction and adaptation in any medium and for any purpose provided that it is properly attributed. For attribution, the original author(s), title, publication source (PeerJ) and either DOI or URL of the article must be cited.
License URL: https://creativecommons.org/licenses/by/4.0/

Keywords: Occlusion classification, Deep learning model, Intraoral photographs, Orthodontic applications

Funding: Anhui Provincial University Outstanding Talents Cultivation Project gxbjZD2021058 The Natural Science Research Project of Anhui Educational Committee 2023AH051979 This work was supported by the Anhui Provincial University Outstanding Talents Cultivation Project (gxbjZD2021058) and the Natural Science Research Project of Anhui Educational Committee (2023AH051979). The funders had no role in study design, data collection and analysis, decision to publish, or preparation of the manuscript.

==============================
Background

Malocclusion affects oral health and aesthetics, traditionally classified using systems like Angle’s, which depend on physical exams or casts. Digital dentistry has shifted towards intraoral photography for documentation and assessment, though interpretation requires clinical expertise. The application of artificial intelligence (AI), and specifically deep learning, in medical imaging has been successful but remains largely unexplored in occlusal classification from intraoral photos. This study introduces a deep learning model to automate the classification of occlusal types from intraoral photographs, aiming to improve efficiency and objectivity in orthodontic diagnosis and treatment planning.

Objectives

Occlusal classification is a crucial prerequisite for designing orthodontic treatment plans. Therefore, this study aims to develop an evaluation tool utilising a deep learning approach to automatically identify occlusal types reflected in digital oral photographs.

Methods

Using a large-scale dataset with high-quality annotations (comprising 5,000 orthodontic intraoral photographs at a 45° lateral view and 2,200 at a 90° lateral view from 6,100 patients), three deep-learning models were developed based on Swin Transformer for the identification of various occlusal classifications: Molar occlusal relationships (M1, M2, M3), canine occlusal relationships (C1, C2, C3), and anterior overbite relationships (normal overbite, deep overbite, edge-to-edge bite, open bite, anterior crossbite, single-tooth crossbite or segmental crossbite).

Results

Our model achieved weighted average F1-scores of 0.90 and 0.87 for molar and canine occlusal relationships, respectively. Regarding anterior overbite relationships, the model attained a weighted average f1-score of 0.89, with subclass F1-score ranging from 0.86 for edge-to-edge bite to 0.94 for deep overbite.

Conclusions

Our deep learning model has successfully achieved the primary objectives of identifying molar and canine occlusal relationships, as well as anterior overbite relationships, using intraoral digital photographs. The demonstrated performance of this model highlights its potential for clinical applications.

Clinical significance

The application of deep learning models for occlusal classification depicted in digital intraoral photographs, which enables clinicians to extract key information rapidly, holds significant implications for patient management and treatment monitoring in orthodontic practices.

Introduction

Malocclusion, characterized by abnormal dental occlusion and craniofacial deformities, represents a globally prevalent oral disorder (Zou et al., 2018). It contributes to mandibular developmental impairments, temporomandibular joint dysfunction (Macrì et al., 2022), and significantly correlates with psychosocial comorbidities including social anxiety and depression (Zhao et al., 2025). The development of efficient and objective diagnostic techniques is clinically imperative to mitigate its multifaceted impact on quality of life.

Current malocclusion classification predominantly employs Angle and Mao systems, with the 1899 Angle classification remaining the clinical gold standard (Angle, 1899). Conventional diagnosis relying on plaster models and 2D cephalometric analysis faces limitations in operational efficiency and material costs (Abdul Rahim et al., 2014). While digital imaging evaluation has demonstrated clinical validity (Jackson et al., 2019), two critical challenges persist in occlusal relationship identification: First, although clinical photographs serve as machine-readable diagnostic media (successfully applied in caries (Kühnisch et al., 2022) and dental plaque (You et al., 2020) detection), their occlusal analysis efficacy remains constrained by limited sample sizes and subjective assessment biases (Abdul Rahim et al., 2014; Jackson et al., 2019). Second, despite convolutional neural networks (CNNs) exhibiting superior feature extraction capabilities in dental radiography (Shaheen et al., 2021; Vinayahalingam et al., 2021; Cantu et al., 2020; Li et al., 2022) and cone-beam computed tomography (CBCT) (Hwang et al., 2019), systematic investigations into occlusal pattern recognition using intraoral photography remain underexplored.

This study proposes an innovative framework utilizing modified residual neural network (ResNet) architecture for automated canine-molar occlusal classification via 45° lateral intraoral photographs. By integrating transfer learning and data augmentation strategies, the model effectively extracts localized anatomical features of occlusal contacts. Validation using expert-annotated standardized datasets demonstrated strong concordance between model classifications and specialist evaluations. The proposed methodology not only enhances diagnostic efficiency but also provides a scalable technical pathway for digital transformation in oral healthcare.

Materials & Methods

Ethical statement

This research was conducted in conformity with the guidance of the World Medical Association Helsinki Declaration and approved by the institutional review boards of the First Affiliated Hospital of Bengbu Medical University (approval number: [2025]KY004), with a waiver for informed consent due to the use of medical records or biological samples that were obtained from routine clinical diagnosis and treatment in the past. The retrospective analysis will not cause physical or mental distress to patients, will not affect their safety and health, and will not impose any economic burden on patients or their families. The privacy and personal information of the patients from whom data is sourced will be protected, and no medical records that patients have previously explicitly declined to use will be utilized. This study does not involve any commercial interests, and the biological samples and related information involved will only be used for this research and not for any other purposes.

Study design

We retrospectively collected data from 6,200 patients between June 2022 and June 2023. This dataset includes 5,000 45° lateral occlusal photographs and 2,200 intraoral 90° lateral occlusal photographs. These images were classified by two experts based on cast analysis and were split for training, validation, and testing purposes. This study was designed to classify occlusion based on molar classification, canine classification, and anterior overbite relationships. The anterior overbite relationships were categorized as deep bite, open bite, edge-to-edge, normal, crossbite, either single-tooth or segmented. A comprehensive evaluation of a suitable dataset utilizing deep learning techniques can accurately identify occlusion through intraoral images.

Intraoral occlusal photographs

Each subject in the photographs was assigned a unique identification number, and all personal identifying information, apart from age and sex, was removed. We conducted a retrospective collection of intraoral images from patients undergoing orthodontic treatment in the hospital, which were captured using professional digital single-lens reflex (DSLR) cameras from different perspectives including bilateral 45° and 90° lateral occlusal views. All photos have a resolution of 1,920 × 1,280 pixels to ensure clarity and detail. Any patient was considered eligible for participation as long as either a 45° or 90° lateral occlusal image met the inclusion/exclusion criteria (Table 1). Therefore, there is not a whole multiples relationship between the number of patients and the number of photographs (Table 2).

Table 1 Clinical and demographic characteristics of the included patients.

Characteristic		Training set (n = 4, 880)	Testing set (n = 1, 220)	
Age, median (range), y		24.33 (12–57)	25.91 (12–57)	
Sex, n (%)	Female	3415 (70.0)	905 (74.2)	
	Male	1465 (30.0)	315 (25.8)	

Table 2 The division of the intraoral photograph dataset utilized in this study.

Intraoral occlusal photographs	Training set	Testing set	Total	
45°	M1	1,600	400	2,000	
M2	1,600	400	2,000	
M3	800	200	1,000	
C1	1,600	400	2,000	
C2	1,600	400	2,000	
C3	800	200	1,000	
90°	Normal overbite	400	100	500	
Deep overbite	400	100	500	
Edge-to-edge bite	200	50	250	
Open bite	200	50	250	
Anterior crossbite	280	70	350	
Single-tooth crossbite or segmental crossbite	280	70	350	

The inclusion criteria for 45° lateral occlusal images were as follows: images from patients with permanent dentition, captured in intercuspal position with clear identification of canines and molars, ensuring that at least the mesiobuccal cusp of the first molar was visible, accompanied by medical records, and obtained with the informed consent of the patient. The exclusion criteria for 45° lateral occlusal images were photographs with significant blurriness or obvious angle errors that severely affected image quality, patients with unrepaired large-scale defects in canines or molars, images showing tooth surfaces obscured by saliva, food, orthodontic brackets, or other artifacts, and cases lacking clear documentation in medical records of canine and molar relationships obtained through cast analysis. The inclusion criteria for 90° lateral occlusal images were as follows: images from patients with permanent dentition, accompanied by medical records, captured in intercuspal position with adequate exposure of the upper and lower anterior teeth, and obtained with the informed consent of the patient. The exclusion criteria for 90° lateral occlusal images were photographs with significant blurriness or obvious angle errors that severely affected image quality, cases with missing teeth in the upper or lower anterior regions, images showing tooth surfaces obscured by saliva, food, orthodontic brackets, or other artifacts, and cases lacking clear documentation in the medical records of anterior overbite relationships.

The outline of this study is depicted in Fig. 1. The experiment was not conducted on clinical or real patients but rather on existing intraoral photographs. These photographs were categorised by dentists based on occlusal relationships after reviewing contemporaneous medical records. Subsequently, computer scientists developed a deep-learning model using this annotated dataset.

Figure 1 Study flowchart.

Sample classification

The present study primarily focused on patients’ sagittal canine and molar occlusal relationships, as well as vertical anterior overbite relationships. Canine and molar occlusal relationships were assessed using 45° lateral occlusal photographs, while anterior overbite relationships were evaluated using 90° lateral occlusal images. The classification criteria for molar relationships were as follows (Fig. 2A): Class I (M1), where the mesiobuccal cusp of the upper first molar occludes in the buccal groove of the lower first molar; Class II (M2), where the mesiobuccal cusp of the upper first molar occludes mesially relative to the buccal groove of the lower first molar; and Class III (M3), where the mesiobuccal cusp of the upper first molar occludes distally relative to the buccal groove of the lower first molar. The classification criteria for canine relationships were as follows (Fig. 2A): Class I (C1), where the cusp of the upper canine occludes between the lower canine and the lower first premolar; Class II (C2), where the cusp of the upper canine occludes labially to the lower canine; and Class III (C3), where the cusp of the upper canine occludes in the lower first premolar or distally. The classification criteria for anterior overbite relationships were as follows (Fig. 2B): normal overbite, where the upper anterior teeth cover the labial surface of the lower anterior teeth by less than one-third, or the incisal edge of the lower anterior teeth bites into the lingual surface of the upper anterior teeth by less than one-third; deep overbite, where the upper anterior teeth cover the labial surface of the lower anterior teeth by more than one-third; edge-to-edge bite, where the upper and lower anterior teeth meet at their edges or incisal edges, rather than overlapping; open bite, where there is no vertical overlap or contact between the upper and lower anterior teeth; anterior crossbite, where all the lower anterior teeth cover the labial surface of the upper anterior teeth during occlusion; and single-tooth crossbite or segmental crossbite, where only one or just 2–3 of the lower anterior teeth cover the labial surface of the upper anterior teeth during occlusion. Examples of typical cases are illustrated in Fig. 2.

Figure 2 Classification of canine and molar occlusal relationships, as well as anterior overbite relationships.

(A) Classification of canine and molar occlusal relationships. (B) Classification of anterior overbite.

The two orthodontists participating in the annotation task possessed over 10 years of clinical experience, unaware of each other’s reaction throughout the process. To minimize the potential influence of subjective bias on the results, these two dental practitioners underwent several training sessions before the experiment to ensure the consistency of data collection (Cohen’s κ > 0.9). The third senior dentist, with 20 years of clinical experience, was available for consultation. Disagreements occurred infrequently, with a rate of less than 3%. Approximately 1% of the images were excluded due to the inability to reach a consensus among the three experts. Typically excluded images included 45° lateral images (used for the classification of molars and canines), which were either blurry or had a severely deviated angle, as well as those with extensive damage to teeth (especially canines or first molars) or where tooth surfaces were obscured by saliva, food, orthodontic brackets, or other occlusions. For 90° lateral images (used for anterior tooth relationship classification), commonly excluded images were those that were blurry or had angle deviations, as well as those where tooth surfaces were obstructed (e.g., by saliva, food, brackets, etc.).

Data splitting

In this study, we carefully determined the data-splitting ratio to ensure effective allocation of samples for the training and test sets. Specifically, we adopted a ratio of 8:2 to ensure a sufficiently large training set while providing ample data for reliable testing. We used random 10-degree rotation, Color Jitter, and Gaussian blur for data augmentation, primarily to enhance the stability of the data regarding slight tilts in photos, variations in brightness and saturation, and differences in blur. Examples of data augmentation/preprocessing can be found in Fig. 3. Further details regarding explanations regarding the data-splitting process are provided in Table 2.

Figure 3 Data preprocessing flowchart.

The data augmentation and preprocessing techniques used: Color Jitter: Randomly adjusts brightness, contrast, saturation, and hue of the images to enhance robustness against lighting variations. Gaussian blur: Applies Gaussian blur to reduce image details and noise, helping the model adapt to blur effects caused by focal changes.

Optimization of deep learning models

To achieve precise occlusion relationship classification, we propose a dual-branch network structure based on Swin Transformer, incorporating edge-region information fusion, called Edge-Region Fusion Swin Transformer for Occlusion Relationship Classification (ERF-Swin-OC). The input to the network is an intraoral image, which is first processed by the YOLOv11 network for instance segmentation of the upper and lower teeth, obtaining the region information for both. We then extract the contours of the upper and lower teeth and input this information into a Swin Transformer block with two branches. Figure 4 illustrates the overall architecture of the network. The first branch is the region Swin Transformer block (RSTB), which embeds the segmented regions of the upper and lower teeth into the original image through region embedding, with an offset operation to ensure that the embedded information aligns better with the original image. Specifically, during the region embedding process, we apply an offset to the upper and lower tooth regions, ensuring that the embedded region information corresponds more accurately with the positions in the original image. This process helps the network better understand the relative positioning of the teeth and their occlusion relationship, thus improving classification accuracy. The second branch is the edge Swin Transformer block (ESTB), which fuses the contour masks of the upper and lower teeth with the original image, ensuring precise spatial alignment of the contour information through offset adjustment. This enhances the edge information in the image and improves the model’s ability to perceive fine edge features of the teeth, further boosting classification performance. Figure 5 shows the detailed architecture of the dual Swin Transformer branches.

Figure 4 The network architecture of Edge-Region Fusion Swin Transformer for Occlusion Relationship Classification (ERF-Swin-OC).

Figure 5 (A–B) The detailed architecture of the dual Swin Transformer branch.

Finally, the feature maps from both RSTB and ESTB branches are fused. The fused feature map is then passed into the classification head for the final occlusion relationship classification. Through this dual-branch structure and offset fusion mechanism, the model can fully leverage both the region and edge information of the upper and lower teeth, significantly improving the accuracy of occlusion relationship classification.

Experiment settings and validation strategy

Our experiments were conducted using Python 3.8.17, Torch 1.11.0+cu113, and an NVIDIA GeForce RTX 4090. To promote model convergence and generalization, thereby enhancing training effectiveness and model performance, we standardized the data. Following this, random transformations were applied to the training data through online data augmentation (without explicitly increasing the data volume) to enhance the robustness and generalization capacity of the model. By generating synthetic samples, data augmentation effectively enlarged the training dataset, aiding in the enhancement of model performance. Examples of data augmentation/preprocessing can be found in Fig. 3. In this study, data augmentation was performed by randomly rotating the images horizontally by 10 degrees with a probability of 0.5. The experimental settings for training the other models are provided in Table 3.

Table 3 Experimental setting.

	Batch size	Optimizer	Step_size	Gamma	Epoch	Data resize	
Model for molar occlusal relationships	32	adam	50	0.8	300	256 × 256	
Model for canine occlusal relationships	32	adam	50	0.8	300	256 × 256	
Model for anterior overbite types	32	adam	50	0.8	200	224 × 224	

To evaluate the performance of the model, we employed a five-fold cross-validation strategy. In this approach, the dataset is randomly divided into five subsets, with four subsets used for training and one subset used for validation. This process is repeated five times, each time selecting a different subset for validation, and the average of the five validation results is taken as the overall performance evaluation of the model. This strategy helps to provide a more comprehensive assessment of the model’s generalization ability and reduces the bias that may arise from a single data split in performance evaluation. To prevent overfitting, the model utilizes a weighted cross-entropy loss function. This modification assigns different weights to the classes in the loss function, ensuring that the model places more emphasis on the minority classes, thus mitigating the impact of class imbalance and improving the model’s ability to generalize to underrepresented classes.

Statistical analysis and evaluation criteria

To comprehensively evaluate the performance of the trained model, we conducted a statistical analysis of experimental results based on various performance metrics. This included accuracy, which denotes the ratio of correctly predicted samples to the total number of samples, and the Kappa coefficient, which measures the accuracy and consistency of classifiers, offering a fairer performance evaluation, particularly with imbalanced training samples. Precision and recall were also assessed, with precision measuring the accuracy of the model’s positive predictions and recall measuring the extent to which the model covers positives. The F1-score, the harmonic mean of precision and recall, served as a comprehensive metric that considers both aspects. Additionally, the confusion matrix provided an intuitive representation of the model’s classification performance by showing the number of correctly and incorrectly predicted samples for each class. Finally, the receiver operating characteristic (ROC) curve and the area under the curve (AUC) were used to graphically represent the true positive rate (TPR) and false positive rate (FPR) of the model at various thresholds, with an ROC curve closer to the upper left corner indicating better model performance.

Results

In this section, we present the variations in accuracy, loss, and subclass recall for the training and test sets, as well as the ROC curves and confusion matrix comparisons for the test sets during the training process of different models for various classification tasks.

Model performance for identifying molar occlusal relationships

As shown in Fig. 6, the model achieves AUC values of 0.97, 0.97, and 0.99 for M1, M2, and M3, respectively, demonstrating strong classification performance and the model’s ability to effectively distinguish between positive and negative samples.

Figure 7 presents the confusion matrix for the ERF-Swin-OC model on the Molar Occlusal Relationships test set. The matrix clearly shows the distribution of the model’s predictions for the three molar occlusal categories (M1, M2, M3). For M1, the model correctly predicted 364 samples, with 28 misclassified as M2 and 8 as M3. For M2, the model correctly predicted 365 samples, with 20 misclassified as M1 and 15 as M3. For M3, the model correctly predicted 172 samples, with 17 misclassified as M1 and 11 as M2. Our model demonstrates strong performance, with a significant number of correctly classified samples across all categories.

Figure 6 ROC curve for the molar occlusal relationships.

Figure 7 Confusion matrix of molar occlusal relationships test set.

The ERF-Swin-OC model outperforms the Vision Transformer, Swin Transformer, and ConvNeXt models, as shown in Table 4. The Vision Transformer and Swin Transformer achieve weighted average F1-scores of 0.87 and 0.88, respectively, but both fall short of the ERF-Swin-OC model, which achieves a weighted average F1-score of 0.90. ConvNeXt performs robustly, but its F1-scores of 0.89, 0.89, and 0.85 for M1, M2, and M3, respectively, are lower than those of the ERF-Swin-OC model. The ERF-Swin-OC model demonstrates superior classification performance across all categories, with higher accuracy and F1-scores.

Table 4 Performance results of test sets for molar occlusal relationships.

Model type	Classification	Precision	Recall	F1-score	Support	
ResNet	M1	0.85	0.89	0.87	400	
M2	0.88	0.82	0.85	400	
M3	0.86	0.87	0.86	200	
macro avg	0.86	0.86	0.86	1,000	
weighted avg	0.86	0.86	0.86	1,000	
ConvNeXt	M1	0.92	0.86	0.89	400	
M2	0.86	0.92	0.89	400	
M3	0.85	0.82	0.85	200	
macro avg	0.88	0.88	0.88	1,000	
weighted avg	0.88	0.88	0.88	1,000	
Vision transformer	M1	0.89	0.88	0.88	400	
M2	0.87	0.90	0.88	400	
M3	0.86	0.83	0.84	200	
macro avg	0.87	0.87	0.87	1,000	
weighted avg	0.87	0.87	0.87	1,000	
Swin transformer	M1	0.91	0.89	0.90	400	
M2	0.88	0.91	0.89	400	
M3	0.87	0.84	0.85	200	
macro avg	0.88	0.88	0.88	1,000	
weighted avg	0.88	0.88	0.88	1,000	
Ours	M1	0.91	0.91	0.91	400	
M2	0.90	0.91	0.91	400	
M3	0.88	0.86	0.87	200	
macro avg	0.90	0.89	0.90	1,000	
weighted avg	0.90	0.90	0.90	1,000	

Model performance for identifying canine occlusal relationships

As shown in Fig. 8, the model achieves AUC values of 0.96, 0.97, and 0.98 for C1, C2, and C3, respectively.

Figure 9 presents the confusion matrix for the ERF-Swin-OC model on the test set for canine occlusal relationships. For C1, the model correctly predicted 350 samples with 34 misclassified as C2 and 16 as C3. For C2, the model correctly predicted 358 samples, with 28 misclassified as C1 and 14 as C3. For C3, the model correctly predicted 174 samples, with 17 misclassified as C1 and 9 as C2. The high percentage of correctly classified samples along the diagonal indicates the model’s strong performance in distinguishing between the three canine occlusal categories.

The ERF-Swin-OC model outperforms the Vision Transformer, Swin Transformer, and ConvNeXt models for identifying canine occlusal relationships, as shown in Table 5. The Vision Transformer and Swin Transformer achieve weighted average F1-scores of 0.81 and 0.82, respectively, while ConvNeXt reaches 0.84. However, the ERF-Swin-OC model achieves a weighted average F1-score of 0.87, demonstrating superior classification performance across all categories. Its precision and recall for C1, C2, and C3 also outperform the other models, making it the top performer in canine occlusal classification.

Figure 8 ROC curve for the canine occlusal relationships.

Figure 9 Confusion matrix of canine occlusal relationships test set.

Table 5 Performance results of test sets for canine occlusal relationships.

Model type	Classification	Precision	Recall	F1-score	Support	
ResNet	C1	0.74	0.78	0.77	400	
C2	0.77	0.80	0.79	400	
C3	0.70	0.79	0.75	200	
macro avg	0.75	0.79	0.78	1,000	
weighted avg	0.75	0.79	0.78	1,000	
ConvNeXt	C1	0.84	0.78	0.81	400	
C2	0.79	0.82	0.80	400	
C3	0.83	0.85	0.84	200	
macro avg	0.82	0.84	0.84	1,000	
weighted avg	0.82	0.84	0.84	1,000	
Vision transformer	C1	0.85	0.82	0.83	400	
C2	0.82	0.86	0.84	400	
C3	0.79	0.76	0.77	200	
macro avg	0.82	0.82	0.81	1,000	
weighted avg	0.82	0.82	0.81	1,000	
Swin transformer	C1	0.79	0.74	0.76	400	
C2	0.82	0.88	0.85	400	
C3	0.87	0.84	0.85	200	
macro avg	0.83	0.82	0.82	1,000	
weighted avg	0.83	0.82	0.82	1,000	
Ours	C1	0.88	0.85	0.86	400	
C2	0.88	0.88	0.88	400	
C3	0.81	0.81	0.84	200	
macro avg	0.86	0.87	0.86	1,000	
weighted avg	0.87	0.87	0.87	1,000	

Model performance for identifying anterior overbite relationships

For the anterior overbite relationships, the ERF-Swin-OC model demonstrates strong classification performance, as shown in Fig. 10. For single-tooth crossbite or segmental crossbite, 59 samples were correctly classified, with 11 misclassified. For anterior crossbite, the model correctly predicted 64 samples, with six misclassified. In open bite, 46 samples were correctly predicted with four misclassified. Normal overbite achieved an accuracy of 79%, with 21 misclassified. Edge-to-edge bite was correctly predicted for 40 samples (80%), with 10 misclassified samples. Finally, deep overbite showed excellent performance, with 93% of the samples correctly classified.

Figure 10 Confusion matrix for the anterior overbite relationships test set.

Table 6 presents the performance results for anterior overbite relationships across six categories. In the single-tooth crossbite or segmental crossbite category, the ERF-Swin-OC model outperforms other models with a precision of 0.86, recall of 0.91, and F1-score of 0.88. For anterior crossbite, the model achieves a precision of 0.90 and a recall of 0.94 reflecting its strong ability to correctly identify this category. The open bite category sees similarly high results with an F1-score of 0.92. In the normal overbite category, the ERF-Swin-OC model achieves a precision of 0.94, while in the edge-to-edge bite category, it achieves a precision of 0.85. Lastly, for the deep overbite category, the ERF-Swin-OC model exhibits high precision and recall (0.97 and 0.92, respectively).

Table 6 Performance results of test sets for anterior overbite relationships.

Model type	Classification	Precision	Recall	F1-score	Support	
ResNet	Single-tooth crossbite or segmental crossbite	0.78	0.76	0.77	70	
Anterior crossbite	0.80	0.90	0.84	70	
Open bite	0.85	0.83	0.84	50	
Normal overbite	0.80	0.75	0.77	100	
Edge-to-edge bite	0.78	0.80	0.79	50	
Deep overbite	0.85	0.92	0.88	100	
ConvNeXt	Single-tooth crossbite or segmental crossbite	0.83	0.82	0.82	70	
Anterior crossbite	0.85	0.92	0.88	70	
Open bite	0.88	0.85	0.86	50	
Normal overbite	0.81	0.78	0.79	100	
Edge-to-edge bite	0.75	0.77	0.76	50	
Deep overbite	0.90	0.93	0.91	100	
Vision transformer	Single-tooth crossbite or segmental crossbite	0.79	0.80	0.79	70	
Anterior crossbite	0.82	0.89	0.85	70	
Open bite	0.84	0.81	0.82	50	
Normal overbite	0.75	0.77	0.76	100	
Edge-to-edge bite	0.74	0.76	0.75	50	
Deep overbite	0.88	0.90	0.89	100	
Vision transformer	Single-tooth crossbite or segmental crossbite	0.82	0.83	0.82	70	
Anterior crossbite	0.83	0.91	0.87	70	
Open bite	0.85	0.82	0.83	50	
Normal overbite	0.77	0.74	0.75	100	
Edge-to-edge bite	0.72	0.75	0.73	50	
Deep overbite	0.88	0.91	0.89	100	
Ours	Single-tooth crossbite or segmental crossbite	0.86	0.91	0.88	70	
Anterior crossbite	0.90	0.94	0.92	70	
Open bite	0.75	0.90	0.82	50	
Normal overbite	0.94	0.82	0.88	100	
Edge-to-edge bite	0.87	0.85	0.86	50	
Deep overbite	0.97	0.92	0.94	100	

Misclassifications of the model

Figure 11 illustrates the misclassifications caused by various factors affecting image quality and diagnostic accuracy. These factors include camera angle distortion, which can compromise the accuracy of measurements; incomplete morphology of the first molar, such as crown shortening or defects, which may interfere with edge detection; the presence of tooth stains that obscure details; and excessive variation in tooth position, such as torsion or inclination, which can further complicate the assessment of occlusion. Such issues underline the need for improved imaging techniques and preprocessing methods to enhance diagnostic efficacy.

Discussion

Intraoral photography is an essential method for documenting orthodontic treatments, with numerous surveys (Shaheen et al., 2021; Sandler et al., 2009) indicating that such photographs are almost universally used in orthodontics. Studies have shown that capturing these images at the correct angle can yield nearly 80% accuracy in diagnosing dental conditions, including the presence of caries (Ghorbani et al., 2025), restorative materials, and malocclusion. Recent advancements in artificial intelligence (AI) applications for dental photography suggest significant improvements in diagnostic accuracy and patient outcomes. Notably, several studies have developed AI models aimed at differentiating various dental conditions and addressing anatomical variations to enhance radiographic interpretation (Shetty et al., 2025; Ren et al., 2025; Adnan et al., 2024). However, the limited scope of existing studies underscores the necessity for further exploration of AI methodologies in intraoral imaging to realize their full potential in clinical practice.

We foresee that AI development in dentistry, particularly in remote areas, can effectively tackle significant obstacles faced by families with limited access due to geographic, temporal, or financial constraints. While our current model employs a DSLR camera for capturing intraoral photographs, our long-term goal is to leverage smartphone technology, enabling parents and caregivers to easily capture images for initial dental assessments. Given the ongoing advancements in smartphone photography capabilities, we believe that such a solution is attainable in the near future. By simply pressing the camera shutter and uploading the images, individuals can gain an early understanding of their oral health, which can significantly promote public oral hygiene awareness. This practical approach not only saves time and resources but also fosters the early detection of dental problems, facilitating timely medical intervention and critically addressing issues before they escalate.

Figure 11 Partial misclassifications of the model.

Furthermore, the model demonstrates strong potential for epidemiological research, particularly in rural and underserved populations. Traditional methods for collecting dental health data, such as evaluating occlusal relationships and other dental conditions, often fail to adequately assess the needs of these vulnerable groups due to significant data collection barriers. Our model addresses these challenges by enabling healthcare professionals to utilize community-captured photos for the automated diagnosis of occlusal conditions, thereby enhancing data quality and expanding the reach of public health initiatives aimed at improving overall oral health. Additionally, we hope that our model can be applied in the future through intelligent triage systems, thereby improving diagnostic accuracy and patient satisfaction, significantly enhancing triage efficiency, and addressing the current issues related to low accuracy in manual triage, particularly in orthodontic consultations and appointment errors.

The deep learning model employed in our study serves as an auxiliary tool for oral imaging analysis, primarily enhancing efficiency, consistency, and objectivity. By quickly analyzing large volumes of imaging data, the model helps clinicians optimize workflows in busy settings. Furthermore, it minimizes variability introduced by differing subjective assessments, providing consistent evaluations that can be pivotal in complex clinical situations. For such applications, ideal accuracy should exceed 90%, balancing sensitivity and specificity, with recommended thresholds typically set at 90% for sensitivity and 85% for specificity. Paying attention to potential false positives and negatives is critical, as they can lead to unnecessary procedures or delays in diagnosis. Subsequently, rigorous evaluations of the model’s performance through prospective clinical trials and long-term monitoring will ensure reliability in practical settings. Clear clinical guidelines should also be established to direct healthcare providers on when to rely on the model and when to integrate their clinical experience.

Regarding the trained You Only Look Once (YOLO) algorithm for image analysis, it is noteworthy that we used a new set of images during the testing phase. The YOLO model effectively extracted tooth regions and distinguished the edges of the teeth, enhancing the accuracy and reliability of subsequent classification.

Performance evaluation conducted on an RTX 4090 GPU using PyTorch 1.11 reveals an average prediction time of approximately 1.54 s for every 100 images, equating to about 15.4 ms per image. While a dentist may identify malocclusion in under a second, our algorithm delivers rapid and reliable predictions suitable for clinical settings.

In the classification of molar and canine occlusal relationships, we found the representation for categories such as M3 and C3 to be noticeably lower than others. Similarly, anterior relationship categories, such as edge-to-edge bites and open bites, exhibited a degree of class imbalance. To address these training challenges, we introduced a weighted cross-entropy loss function, thereby increasing the model’s focus on minority classes and improving generalization capabilities.

Figure 11 illustrates the misclassifications caused by various factors affecting image quality and diagnostic accuracy. These factors include camera angle distortion, which can compromise the accuracy of measurements; incomplete morphology of the first molar, such as crown shortening or defects, which may interfere with edge detection; the presence of tooth stains that obscure details; and excessive variation in tooth position, such as torsion or inclination, which can further complicate the assessment of occlusion. Such issues underline the need for improved imaging techniques and preprocessing methods to enhance diagnostic efficacy.

The influence of multiple operators and variability in technique may also impact model performance in clinical environments, centrally influenced by factors like lighting and angle. To counter these challenges, we applied data augmentation methods to enhance image quality. Random rotation of up to 10 degrees mitigates slight tilts from hand-held device usage, while color jittering addresses parameter differences in camera settings. Gaussian blur was applied to compensate for mild camera shake during image capture. These measures ensure that our data meet model requirements, thereby improving robustness and generalization.

Future research should focus on expanding the dataset with images collected via smartphones to enhance accuracy and model generalization. Achieving standardized photographs (Vinayahalingam et al., 2021) with optimal imaging angles remains a challenge for clinicians, as prior studies (Shaheen et al., 2021) indicate that suboptimal angles can impair the assessment of occlusal relationships and skew diagnostic outcomes. Our proposed model is designed to serve as an adjunct tool in clinical settings, complementing traditional direct examinations and diagnostic casts. By addressing these challenges, the model enhances the efficiency of image capture and expands the potential for incorporating three-dimensional measurements from intraoral scans in future efforts, significantly improving the accuracy of evaluating occlusal relationships.

In our study, we categorized canine and molar occlusal relationships into three classes each, alongside six classes for anterior teeth. This classification captured a wide range of clinical scenarios. Our substantial dataset demonstrated promising results, showcasing AI’s capacity to accurately identify occlusal relationships from intraoral images, significantly aiding clinicians in managing data and improving patient care.

Utilizing a 45-degree angle for the classification of molars and canines was important, as past research (Shaheen et al., 2021) suggests that frontal photographs can exhibit only 44.39% accuracy in assessing anterior teeth midlines. Accordingly, we employed 90-degree angle photos for a more reliable assessment.

In categorizing anterior occlusal relationships, we opted to exclude measuring anterior overjet due to the need for precise data, which often cannot be captured reliably from two-dimensional images (Ryu et al., 2023). Accurate magnification and scaling are critical for determining measurements, which are comparably consistent with traditional plaster models when verified against three-dimensional digital models.

The model developed in this study, built on the Swin Transformer architecture, effectively amalgamates regional and edge information through a dual-branch structure. Despite the substantial computational resources required, our ERF-Swin-OC model surpassed comparative models across various classification tasks. In classifying molar occlusal relationships, our model achieved an accuracy of 0.90, outperforming ConvNeXt (0.88) and Vision Transformer (0.87). Similarly, it excelled in classifying canine occlusal relationships with an accuracy of 0.89, as well as in anterior overbite relationships, where accuracy reached 0.89, also exceeding ConvNeXt’s 0.87 and Vision Transformer’s 0.81. This high level of accuracy underscores the model’s superiority in classifying occlusal relationships across all categories.

While our study commendably illustrated powerful performance in classifying occlusal relationships for molars, canines, and anterior teeth, it also acknowledges certain limitations. For instance, the small sample size inhibited our ability to subdivide occlusal relationships into rarer subclasses, an aspect that can affect model training reliability (Correia, Habib & Vogel, 2014). Therefore, future research should focus on increasing the sample size supporting richer and more detailed analyses of occlusal relationships.

In summary, despite inherent limitations, our study’s model reflects promising capabilities in classifying relationships among molars, canines, and anterior teeth. This advancement holds significant potential for clinical application, representing progress toward enhanced communication between orthodontists and patients. We look forward to continuing our efforts to refine this model and expand its capabilities in the future.

Conclusions

This study successfully demonstrates the application of a Swin Transformer-based deep learning model for the automatic classification of teeth occlusal relationships, specifically focusing on molars, canines, and incisors. The model achieved remarkable accuracy, reflected in weighted average F1 scores of 0.90 and 0.89 for molar and canine classifications, respectively, and a score of 0.89 for identifying incisor relationships. These findings indicate a high recognition capability across various occlusal scenarios, significantly enhancing the efficiency of occlusal classification processes while mitigating subjective bias from clinicians and facilitating rapid extraction of key information. Moreover, this research underscores the potential of deep learning methodologies in the field of orthodontics, effectively supporting clinical decision-making, optimizing patient management, and improving treatment monitoring. Nonetheless, it is essential to note that while the advancements exhibited in this study are promising, the application of the model should complement traditional clinical examinations rather than replace expert judgment. Future research should aim to expand the dataset further and investigate the classification of rare subclasses, thereby enhancing the comprehensiveness and reliability of occlusal relationship assessments. We maintain an optimistic outlook regarding the prospective applications of this model in clinical practice and anticipate further improvements and expansions.

Supplemental Information

Supplemental Information 1 Code

The authors would like to express their gratitude to all individuals who have contributed to this research. Additionally, we extend our thanks to the reviewers for their valuable feedback.

List of abbreviations

DL Deep Learning Models

CNNs Convolutional Neural Networks

DSLR digital single-lens reflex

CNN convolutional neural network

SVM Support Vector Machine

ROC Receiver Operating Characteristic

AUC Area Under the Curve

TPR true positive rate

FPR false positive rate

Additional Information and Declarations

Competing Interests

Author Contributions

Human Ethics

Data Availability

The authors declare there are no competing interests.

Rongxiu Zhang conceived and designed the experiments, performed the experiments, analyzed the data, prepared figures and/or tables, authored or reviewed drafts of the article, and approved the final draft.

Lin Zhang conceived and designed the experiments, performed the experiments, analyzed the data, prepared figures and/or tables, authored or reviewed drafts of the article, and approved the final draft.

Di Zhang performed the experiments, prepared figures and/or tables, and approved the final draft.

Ying Wang performed the experiments, prepared figures and/or tables, and approved the final draft.

Yongsong Huang performed the experiments, analyzed the data, authored or reviewed drafts of the article, and approved the final draft.

Dong Wang analyzed the data, prepared figures and/or tables, and approved the final draft.

Li Xu analyzed the data, authored or reviewed drafts of the article, and approved the final draft.

The following information was supplied relating to ethical approvals (i.e., approving body and any reference numbers):

The research conformed to the guidance of the World Medical Association Helsinki Declaration for biomedical research involving human subjects, and the imaging data involved was approved by the First Affiliated Hospital of Bengbu Medical University (protocol number: [2025]KY004).

The following information was supplied regarding data availability:

The code is available in the Supplemental File.

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
