# Peer review of "Development and evaluation of a deep learning model for occlusion classification in intraoral photographs"

_PeerJ, doi:10.7717/peerj.20140_

## Round 0.1 · original submission · Minor Revisions

Apologies for the duplicate email, we are troubleshooting your inability to resubmit.

Reviewer 1 ·

Basic reporting

The reporting is clear, and there are some minor typos in the manuscripts, legends, and so on. In terms of literature and background, the manuscript stated quite clearly. The results are sound.

Experimental design

The experimental design is simple and well-designed. In terms of the dataset photos, the requirement of the photos should be stated, not just from DSLR camera but should be in xxx pixels with yyy background. Two experts were used, so do they match in their decision?

Validity of the findings

Using YOLO is easy, and I would like to ask whether this trained algorithm can be used for analyzing new photos. I would also like to ask the time for such a determination - I think the determination of malocclusion is easy for any dentist, which might take <1s to do. How about this algorithm?

Reviewer 2 ·

Basic reporting

The basic reporting is appropriate

Experimental design

The experimental design is sound

Validity of the findings

The findings of the study are valid

Additional comments

The authors can some images of data augmentation, preprocessing in the methodology section.

In the results section some images of misclassification of the images by the model

In the discussion section the authors can some information about recently published studies using ResNet in image classification DOI 10.1038/s41598-025-93250-8.

Overall the manuscript is well presented. Congratulations to the authors for the good work

Reviewer 3 ·

Basic reporting

The manuscript is overall well written and clearly structured. Although the English is mostly clear, there are a few points where the phrasing could be improved for correctness, clarity, flow, and formality. For example:
* “Everyone in the pictures was assigned a unique identification number…” should be revised to: “Each subject in the photographs was assigned a unique identifier…” (Line 102).
* There are some typographic errors. (e.g. Line 342).
* “Datset” should be corrected to “Dataset”. (Figure 1)
Therefore, a round of thorough proofreading would help further refine the manuscript.

The introduction and discussion sections provide a comprehensive rationale for the study, outlining the current challenges in occlusal classification and the need for automated solutions. The background is well supported by relevant literature; however, adding a few more recent references on AI applications in dental photography would strengthen this further.

Additionally, conventional assessment of occlusal relationships in clinical settings is not necessarily difficult or overly time-consuming. A more detailed discussion of how the proposed model could be integrated into real-world clinical workflows would enhance the practical significance of the findings.

Experimental design

- The ground truth labels are critically important for the model training and the validation of the final model, especially when labeling by human. As the authors mention that two experienced orthodontists performed the image labeling, with a Cohen’s kappa of >0.9, and a third expert was consulted in case of disagreement. If possible, the authors should report or discuss about "How frequently disagreements occurred?", "How many images were excluded due to lack of consensus?" or "What types of images are often excluded from experiments?" This will improve transparency and help readers evaluate the reliability of the ground truth labels.
- Please provide a brief explanation for class distribution and discuss how class imbalance may have affected model training.
- If possible, please clarify whether all photographs were taken under standardized protocols and whether multiple operators were involved and discuss how variability in photographic quality and technique may influence model performance, particularly in real-world clinical environments.

Validity of the findings

While the findings are supported by performance metrics, there are several aspects that require further clarification:
- Although confusion matrices are provided, there is no discussion of the types of errors made by the model, nor are any example misclassified images shown or analyzed. More information on this point will provide insight into the model’s limitations and help guide future improvements.
- The authors mention that the model proposed by their study is intended to function as a supplementary tool. Please additionally discuss about the reason and the acceptable threshold of accuracy or reliability from a clinical perspective. How to consider whether the model’s performance is sufficient to support real-world decisions or if it should be used as a secondary tool.

Additional comments

no comment

---

## Round 0.2 · Minor Revisions

Dear Authors,

Thank you for submitting the revised version of your manuscript. We appreciate the improvements you have made.

Before moving forward, we kindly ask you to carefully check the text again for any remaining typos. Additionally, please address the following points:

Figure 1: The English should be corrected.
Figure 3: In the legend, please explain the meaning of Color Jitter and Gaussian Blur.

Reviewer 3 ·

Basic reporting

The revised manuscript demonstrates clear and professional scientific reporting regarding clarity, grammar, phrasing, and minor typographical issues. However, the typos in Figure 1 still exist. Please correct these before final acceptance.

Experimental design

The study presents a clear, well-defined, and scientifically sound experimental framework.

Validity of the findings

The results are valid, well-presented, and supported by robust evaluation metrics.

Additional comments

-

---

## Round 0.3 · accepted · Accept

Manuscript is ready for publication.